# Effect of Innovative Sediment-Based Growing Media on Fruit Quality of Wild Strawberry (*Fragaria vesca* L.)

Edgardo Giordani [1], Lorenzo Bini [1,*], Daniele Bonetti [2], William Antonio Petrucci [1], Grazia Masciandaro [3], Giacomo Chini [4] and Stefania Nin [2]

1. Department of Agriculture, Food, Environment, and Forestry (DAGRI), University of Florence, Viale delle Idee 30, 50019 Sesto Fiorentino, FI, Italy; edgardo.giordani@unifi.it (E.G.); williamantonio.petrucci@unifi.it (W.A.P.)
2. CREA Research Centre for Vegetables and Ornamental Crops, Via dei Fiori 8, 51017 Pescia, PT, Italy; daniele.bonetti@crea.gov.it (D.B.); stefania.nin@crea.gov.it (S.N.)
3. Research Institute on Terrestrial Ecosystems, National Research Council of Italy (CNR-IRET), Via Moruzzi 1, 56124 Pisa, PI, Italy; grazia.masciandaro@cnr.it
4. Carbonsink s.r.l., Via Madonna del Piano 6, 50019 Sesto Fiorentino, FI, Italy; giacomochini94@gmail.com
* Correspondence: lorenzo.bini@unifi.it

**Abstract:** The aim of this research was to assess the quality attributes of the fruits of wild strawberry (*Fragaria vesca*) "Regina delle Valli" plants cultivated in pots in remediated-sediment-based growing media (GM). For this purpose, phytoremediated and landfarmed dredged sediment from Leghorn Harbor (Italy) was mixed into a peat-based commercial substrate at different volume/volume percentages (0, 50 and 100%), and the plants were grown under greenhouse conditions with two different water regimes (WR1 = 950 cc/day; WR2 = 530 cc/day). The fruit quality parameters were differentially affected by the main factors (GM and WR) and their combinations. The fruits obtained from the substrate richest in treated sediment and with the higher water regime (TS100-WR1) showed the highest content of sugars (4056, 5256 and 5178 mg/100 g FW of sucrose, glucose and fructose, respectively), total soluble solids (16 °Brix) and organic acids (30, 490 and 2300 mg/100 g FW of ascorbic, malic and citric acid, respectively). Neither the polyphenol content nor the DPPH radical scavenging activity were significantly affected by the treatments, although the TS100-WR1 plants showed the highest DPPH value (an inhibition of 0.71%). Among the analyzed organic contaminants, only total heavy hydrocarbons (C10–C40) were found in fruits from TS50 and TS100 at very low values, while heavy metals were not detected. The PCA statistical multivariate analysis performed on the visual, olfactory, chewing and tasting aspects of a sensorial evaluation clearly showed that the substrate's chemical–physical properties exerted a relevant influence on the fruit samples, while the irrigation regimes did not affect significantly fruit quality parameters. A number of highly correlated soil/fruit parameters were found. The remediated sediment proved to be a valid alternative to peat for wild strawberry production, even at the highest concentration.

**Keywords:** soilless cultivation; peat replacement; maritime port; sensory analysis

## 1. Introduction

Maritime port dredging is a compulsory activity to guarantee the transit of ships. It consists of a massive movement of polluted sediments from the sea to the open air and thus has negative effects on the environment [1–3]. The potential reuse of port-dredged sediments is restricted by European legislation, which defines them as contaminated residues since, in general terms, they are rich in heavy metals and polycyclic aromatic hydrocarbons (PAHs) [4,5]. In recent years, multidisciplinary scientific research has demonstrated the possibility of using dredged sediments from rivers, lakes and maritime ports as new substrates for plant cultivation after their remediation [6]. These novel substrates may represent a suitable product for agricultural production in order to partially or totally

replace the current peat-based commercial substrates, which are characterized by the high environmental impact of peat extraction and transport [7,8]. Several recently published research papers show that alternative growing substrates obtained in the frame of a circular economy approach can be suitable for the production of edible crops, such as lettuce, strawberry and pomegranate [8–10].

The production and commerce of small berries represent a valid sustainable alternative income for growers located in marginal areas. Among these berries, the fresh fruits of wild strawberry (*Fragaria vesca* L.) are unique for their content of bioactive phenolics and for their scent, which makes them particularly desirable for use in making pastry [11]. Nonetheless, the fruits of wild strawberries are frequently hard to locate in close proximity to the destination of their intended consumption due to their limited suitability for post-harvest storage and short shelf life. For this reason, economically sustainable soilless production systems are being improved for the commercial cultivation of *F. vesca* via the adoption of new substrates for pot cultivation. This approach offers the possibility to obtain more crops per year, enhance off-season production, and reduce harvest labor costs while enabling easier harvest handling and cleaner fruit production and eliminating the need for soil sterilization [12]. It is estimated that the total Italian acreage of land under *F. vesca* cultivation is about 100 hectares, amounting to a total production of about 400 t per year. Successful production has mostly been developed in the north of Italy, especially in the alpine zones of Piedmont and Trentino, between 400 and 1200 m a.s.l. In the middle west of the country, the most favorable soil and climatic conditions for growing local ecotypes similar to "Regina delle Valli" are found in the provinces of Salerno and Avellino [13] and also in Tuscany [12]. Other countries, such as the USA and Poland, are improving the cultivation techniques and enhancing the production of *F. vesca* cultivars [14,15].

Prior studies have established that the composition of the growing media and irrigation practices can impact the yield, fruit quality and various morphological, phenological and organoleptic parameters of wild strawberries [9,11]. In this context, a previous work [16], which was based on the same experiment reported below, focused on the productivity of *F. vesca* "Regina delle Valli", demonstrating that the growth and yield of plants cultivated in sediment-based growing media were comparable or superior to those grown in commercial peat. Specifically, in this research paper, we investigate how remediated-sediment-based growing substrates and various water regimes affect the quality traits, such as nutraceutical properties and sensorial parameters, of "Regina delle Valli" fruit.

## 2. Materials and Methods

### 2.1. Growing Media Components, Plant Material and Cultivation

Marine sediment dredged from Leghorn Port (Central Italy, 43°33′25″ N, 10°17′39″ E) was phytoremediated and subjected to three years of landfarming [17]. The cultivation trials were performed at the Az. Agricola Franceschini (Central Italy, 43°51′32.1″ N 10°41′11.3″ E) during early Spring 2020 and late Autumn 2021. Three different growing media (GMs) were adopted for the trial: (i) a commercial peat-based substrate used as a control (TS0); (ii) a mixed substrate containing the commercial peat-based substrate and the remediated sediment at a ratio of 50:50 $v/v$ (TS50); (iii) the remediated sediment alone (TS100). The physico-chemical characterization of the three substrates was performed according to Macci et al. [17] before transplanting (Table 1). Wild strawberry commercial certified plants (*F. vesca*) cv. "Regina delle Valli" were conventionally cultivated in 50 L rectangular plastic pots under controlled greenhouse conditions and two water regimes per pot (WR1 = 960 cc/day and WR2 = 530 cc/day), with a total of 90 plants for each GM×WR combination (3 growing media × 2 water regimes, replicated in 3 blocks, each consisting of 5 plants). Drip irrigation was used to control the water supply, and all plants received water from the same reservoir. Two distinct line drippers were used to manage the water supply; each had six drip emitters per pot at a flow rate of 80 cc min$^{-1}$, and the irrigation time ranged from 1.1 to 2 min per day. The pH level of the irrigation water was maintained between 6.0 and 6.5. The plants were fertigated once per week with Universol$^{®®}$ water

soluble fertilizer, NPK 15:7:30 + 3.0 MgO + TE, in the range of 200 g per plant in the periods May–June and September–April and 400 g per plant from July to August. Growing media TS0, TS50 and TS100, with the respective WRs, were analyzed after plant cultivation to obtain a chemico-physical analysis according to Macci et al. [17] (Table S1).

**Table 1.** Physico-chemical properties of container media before the transplanting of *F. vesca* cv "Regina delle Valli".

| Parameters | TS0 | TS50 | TS100 |
|:---:|:---:|:---:|:---:|
| Texture | n.a. | n.a. | Sandy/Loam |
| Bulk density (g cm$^3$) | 0.19 | 1.01 | 1.52 |
| Porosity (%) | 95 | 89 | 76 |
| Water capacity (%) | 81 | 76 | 64 |
| Air capacity (%) | 14 | 13 | 11 |
| Easily available water (%) | 42 | 29 | 20 |
| pH | 5.6 | 7.8 | 8.6 |
| EC (dS m$^{-1}$) | 0.68 | 0.52 | 0.28 |
| TN (%) | 0.63 | 0.15 | 0.09 |
| TOC (%) | 7.96 | 3.52 | 1.34 |
| TP (g Kg$^{-1}$) | 0.30 | 0.40 | 0.38 |
| $NO_3^-$ (mg Kg$^{-1}$) | 42.9 | 28.5 | 45.7 |
| $NH_4^+$ (mg Kg$^{-1}$) | 11.40 | 6.73 | 0.61 |
| **Minerals (g Kg$^{-1}$)** | | | |
| Pb | 0.02 | 0.04 | 0.05 |
| Fe | 5.00 | 15.07 | 17.41 |
| Cr | 0.01 | 0.04 | 0.05 |
| Cu | 0.01 | 0.04 | 0.05 |
| Zn | 0.02 | 0.16 | 0.17 |
| Ni | 0.01 | 0.04 | 0.04 |
| Mn | 0.18 | 0.28 | 0.28 |
| Mg | 1.99 | 5.14 | 5.76 |
| Cd | - | - | - |
| K | 5.69 | 3.04 | 1.95 |
| Ca | 4.82 | 25.73 | 24.04 |
| Na | 1.32 | 1.48 | 1.35 |
| **Enzimatic activity (μmol/g$^{-1}$ h$^{-1}$)** | | | |
| ß-Glucosidase | 132 | 91 | 90 |
| Phophatase | 1446 | 476 | 147 |
| Aryl Sulfatase | 0 | 40 | 70 |
| Butyrate esterase | 114 | 412 | 574 |

Electrical conductivity; total nitrogen (TN); total organic carbon (TOC); total phosphorus (TP); limit of detection (0.01 ppm) (LOD).

## 2.2. Leaf Sampling and Pigment Analysis

A pigment analysis (chlorophylls and carotenoids) was performed during a period of full vegetative activity (June 2021) as an indicator of the plants' cellular metabolic state. Leaf samples (5–6 fully expanded leaves collected from the middle portion of the plants of each growing medium × water regime replicate) were used for the preparation of leaf discs, and pigment extraction was carried out according to the Lichtentahler and Buschmann [18] technique. The absorbances of chlorophyll a, chlorophyll b and carotenoids were read with a spectrophotometer (Thermo Evolution 300 UV-Visible Spectrophotometer) at 665.2 nm, 652.4 nm and 470 nm, respectively.

## 2.3. Fruit Sampling and Quality Assessment

Ripened, marketable fruits, characterized by a red color all over the fruit, were collected in suitable amounts and adequately conditioned for nutraceutical, elemental and contaminant analyses. All analyses were carried out in analytical triple. Fruit samples from each plant per block belonging to the same GM×WR treatment were collected, mixed and immediately weighed and freeze-dried.

### 2.3.1. Nutritional and Nutraceutical Analysis

### Determination of Sugars

For the quantification of sugar via the HPAEC-PAD system, 5 g of frozen, dried strawberry was homogenized in 25 mL of water at a temperature of 30 °C. The resulting mixture was then extracted via sonication for a duration of 20 min. Thereafter, the suspensions were centrifuged, and the supernatant was filtered and injected. Carbohydrate analyses were performed using a Dionex system (Dionex Corp., Sunnyvale, CA, USA) equipped with a pulsed amperometric detector (model ED40), using an injection loop of 10 µL and a Dionex CarboPac PA100 analytical column (250-mm × 4-mm i.d.). In addition, a guard CarboPac PA100 column (50-mm × 4-mm i.d.) was used to optimize the identification and quantification. The eluent used had a flow rate of 0.5 mL min$^{-1}$ and consisted of 40 mM sodium hydroxide and 1 mM barium acetate. The eluent was prepared with carbonate-free 50 wt% NaOH using degassed water continuously sparged with purified $N_2$. The detection potential at the gold working electrode was EDET +0.10 V vs. Ag | AgCl, according to the literature [19].

### Total Soluble Solids (TSS)

Samples of 5 g of strawberry were homogenized with an ULTRA-TURRAX (VWR International, Radnor, PA, USA) homogenizer at 20,000 rpm for 5 min and centrifuged at 5000 rpm for 30 min, and the supernatant was measured with a hand refractometer (Atago N1-Atago Co., Tokyo, Japan).

### Determination of Organic Acids

For sample preparation, the strawberry extract was diluted 1:100 in acidic water (0.5% formic acid), and succinic acid (d4) was added as internal standard whose final concentration was 5 ng µL$^{-1}$. The analysis was performed using an HPLC system and X Terra RP18 Column (particle size 5 µm; 4.6 × 150 mm; at 18 °C). The mobile phases were acidic $H_2O$ (0.5% formic acid) (phase A) and acidic MeOH (0.5% formic acid + IPA 10%) (phase B) at variable flow rates. Detection was performed with a triple quadrupole SCIEX 4000 Qtrap with a Turbo V ESI source, using MRM in negative mode.

### Determination of Organic Phenolics

The extracts' total phenolic contents were quantified via the Folin–Ciocalteu method according to the procedures used by Duarte-Almeida et al. [20]. Accordingly, 0.1–0.5 mL of extract was added to 0.1 mL of Folin–Ciocalteu with 6 mL of distilled water. Then, 2 mL of sodium carbonate (15%) and 10 mL of distilled water were added and mixed at room temperature. Incubation was performed in the dark at 37 °C for 2 h, and the absorbance

was measured at 750 nm. Gallic acid (Sigma-Aldrich, St. Louis, MO, USA) was used as a standard, and the total phenolic contents were expressed as milligrams of gallic acid equivalents per 100 g of fresh strawberry (mg GAE/100 g FW).

Determination of 2,2-Diphenyl-1-Picrylhydrazyl (DPPH) Radical-Scavenging Activity

Antioxidant activity was assessed in terms of the hydrogen-donating or radical-scavenging abilities of 2,2-diphenyl-1-picrylhydrazyl (DPPH), according to the method used by Chang et al. [21]. A total of 5 g of each composite strawberry sample was mixed with 1 mL of 1 mM freshly prepared DPPH ethanolic solution and left to stand for 30 min prior to being spectrophotometric detection at 517 nm using a Spectrophotometer UV-1900 Shimadzu. The percentage of DPPH scavenging activity (%) was expressed by $((\Delta A517$ of control $- \Delta A517$ of sample)$/\Delta A517$ of control) $\times 100$.

### 2.3.2. Elemental and Organic Contaminant Analyses

Nutrients, heavy metals and organic micropollutants were analyzed by an external accredited laboratory (LabAnalysis, accessed on 26 May 2022), following the official analytical protocol standards adopted in Europe and Italy. For the determination of nutrient (Ca, K, Mg, Mn, Na and P) and heavy metal (Cd, Cr, Cu, Fe, Ni, Pb and Zn) concentrations, 500 mg of each pulverized, dried fruit was mineralized via microwave-assisted acid digestion (Ethos 1, Milestone s.r.l., Sorisole, Bergamo, Italy), using 25 mL of $H_2O_2$:$HNO_3$ 1:3 (*v/v*), and analyzed by ICP. Blanks were prepared for each analytical batch to account for potential metal contamination of the reagents and reactors [22–24]. The total nitrogen was analyzed according to the [25], while hydrocarbons with carbon numbers in the range of 10–40 ($C_{10–40}$) [26] and polycyclic aromatic hydrocarbons (PAHs = anthracene, benzo(a)anthracene, benzo(b)fluoranthene, benzo(g,h,i)perylene, benzo(k)fluoranthene, chrysene, dibenz(a,h)anthracene, phenanthrene, fluoranthene and fluorene) were analyzed via HLPC-FLD, according to [27].

### 2.3.3. Sensory Evaluation

Fresh, ripened fruits, a total of approximately 60 units for each GM×WR treatment, were collected 3 h before the sensory analysis and used for the panel test. The fruits were presented to the panelists via randomized, three-digit-coded blind samples placed in non-transparent glass containers and were evaluated at each testing session. Samples were presented to the assessors in two replicates in an air-conditioned room set at 24 °C. Mineral water and crackers were provided for palate cleansing. The panelists (n = 10), trained in sensory lexicon and methodology (according to the ISO 1993), with previous experience in assessing fruit products, evaluated all the samples twice in a random sequence. The list of sensory attributes included 8 groups of descriptors chosen directly by the assessors, namely, color intensity, hue and uniformity, fruit roundness and integrity as visual attributes; smell intensity, fruity, floral, herbaceous, undergrowth, spicy, vinous and vanilla smells and off-flavors as olfactory attributes of the entire fruit; fruit consistency, graininess, viscosity, astringency, sweetness, sourness, bitterness during mastication; flavor intensity, fruity, floral, herbaceous, undergrowth, spicy, vinous, and vanilla and off-flavor aftertastes; and general satisfaction after the trial. The same panelists evaluated the attributes of fruits from all cultivars. The sample evaluation was performed using a score range of 1 to 9 for all descriptors (1 = none; 9 = high intensity).

### 2.4. Data Analysis

The data were analyzed by 2-factorial ANOVA, followed by Duncan's multiple range comparison test at the level of significance of 95% ($p < 0.05$), using SPSS 20 (Chicago, IL, USA). Regarding the sensorial analysis, data were processed and evaluated with PanelCheck V1.4.2 software "www.panelcheck.com (accessed on 23 November 2022)". Particularly, the response variables were first tested for normality and variance heterogeneity. Sub-

sequently, a spider plot and a principal component analysis (PCA) were generated and carried out, first for each sensorial set and then considering all the parameters simultaneously.

Moreover, a multivariate analysis was performed considering all plant variables for each GM. As an initial step, a correlation plot of these traits was generated with R-package corrplot 0.92 [28], according to the Pearson correlation coefficient, to reduce the number of parameters analyzed. A PCA biplot was achieved for the selected plant variables using the FactoMineR 2.7 [29] and factoextra 1.0.7 [30] packages in R-project. Finally, a heatmap chart was generated to graphically display the relationship between all soil properties (measured at the end of the cultivation) and all plant parameters using the R-package heatmap 1.0.12 [31].

## 3. Results

### 3.1. Leaf Pigment Analysis

Regarding the pigment contents observed in plants cultivated in different growing media and irrigation regimes, the average values and the significance of the single factors and their combination are reported in Table 2. Differences in the average values were statistically relevant for carotenoids. Of these values, TS100 showed the lowest value (12.2 ± 2.8 µg/mg FW), approximately half of the values registered for TS0 and TS50, while no significant variations were detected between the two water regimes. In addition, the variations noticed for the overall average values of Chl a, Chl b, Chl a + Chl b and Chl a/Chl b among the different treatments and their interactions were statistically negligible (Table 2).

**Table 2.** Carotenoid and chlorophyll a and b contents in leaves of wild strawberry (*Fragaria vesca*) plants grown in three growing media with increasing TS under different irrigation regimes. Mean separation within columns by Duncan's multiple range comparison test. Means and standard deviations followed by different letters are significantly different; ** significant at $p < 0.01$, ns = not significant.

| Factor | Carotenoids (µg/mg FW) | Chl a (µg/mg FW) | Chl b (µg/mg FW) | Chl a + Chl b (µg/mg FW) | Chl a/Chl b (µg/mg FW) |
|---|---|---|---|---|---|
| **GM** | | | | | |
| TS0 | 28.80 ± 3.1 a | 1.23 ± 0.19 ns | 0.58 ± 0.12 ns | 1.80 ± 0.26 ns | 2.20 ± 0.45 ns |
| TS50 | 23.98 ± 2.7 a | 1.25 ± 0.11 ns | 0.57 ± 0.11 ns | 1.82 ± 0.16 ns | 2.27 ± 0.45 ns |
| TS100 | 12.20 ± 2.8 b | 1.09 ± 0.20 ns | 0.71 ± 0.15 ns | 1.80 ± 0.27 ns | 1.68 ± 0.34 ns |
| **WR** | | | | | |
| WR1 | 19.60 ± 8.19 ns | 1.21 ± 0.10 ns | 0.67 ± 0.13 ns | 1.89 ± 0.20 ns | 1.96 ± 0.39 ns |
| WR2 | 23.72 ± 9.50 ns | 1.17 ± 0.22 ns | 0.56 ± 0.09 ns | 1.73 ± 0.23 ns | 2.14 ± 0.43 ns |
| **GM × WR** | | | | | |
| TS0-WR1 | 28.11 ± 2.9 a | 1.24 ± 0.14 ns | 0.58 ± 0.13 ns | 1.82 ± 0.21 ns | 2.16 ± 0.36 ns |
| TS0-WR2 | 29.49 ± 3.3 a | 1.22 ± 0.23 ns | 0.57 ± 0.12 ns | 1.79 ± 0.22 ns | 2.24 ± 0.28 ns |
| TS50-WR1 | 18.95 ± 2.7 b | 1.19 ± 0.16 ns | 0.62 ± 0.14 ns | 1.81 ± 0.14 ns | 1.97 ± 0.44 ns |
| TS50-WR2 | 29.01 ± 2.9 a | 1.32 ± 0.28 ns | 0.52 ± 0.10 ns | 1.84 ± 0.26 ns | 2.56 ± 0.58 ns |
| TS100-WR1 | 11.75 ± 2.1 c | 1.22 ± 0.21 ns | 0.81 ± 0.23 ns | 2.03 ± 0.31 ns | 1.74 ± 0.38 ns |
| TS100-WR2 | 12.65 ± 2.4 c | 0.97 ± 0.12 ns | 0.60 ± 0.09 ns | 1.56 ± 0.23 ns | 1.62 ± 0.39 ns |
| **Significance** | | | | | |
| GM | ** | ns | ns | ns | ns |
| WR | ns | ns | ns | ns | ns |
| GM × WR | ** | ns | ns | ns | ns |

### 3.2. Nutritional and Nutraceutical Analysis

Statistically significant differences were found with respect to the contents of both sugars and organic acids, although the sugar content was quite homogeneous in the fruits obtained from the plants grown in all different treatments (Table 3). Specifically, sucrose (overall mean 3546 ± 290 mg/100 g FW), glucose (overall mean 4551 ± 359 mg/100 g FW) and fructose (overall mean 4574 ± 297 mg/100 g FW) showed very similar amounts among the fruits obtained from different growing media and from the two irrigation regimes except for TS100-WR1, which revealed the highest values (~1.10/15-fold greater than the average). The lowest TSS value was observed in the fruits from TS50-WR1 (12.4 ± 0.5 ° Brix), while the highest was observed in the TS100-WR1 fruits (16 ± 0.7 °Brix), leading to an overall average of 13.7 ± 1 °Brix. Furthermore, the citric acid concentration ranged from 1000 ± 87 mg/100 g FW in TS100-WR2 fruits to 2300 ± 98 mg/100 g FW in TS100-WR1 fruits; this latter treatment also displayed the highest significant values of malic and ascorbic acids (490 ± 43 and 30 ± 1.2 mg/100 g FW, respectively).

**Table 3.** Sugar and organic acid contents in the fruits of wild strawberry (*F. vesca* cv "Regina delle Valli") cultivated in three growing media with increasing amounts of treated port sediment (TS0, TS50 and TS100) under two irrigation regimes (WR1 and WR2). Means and standard deviations followed by different letters are significantly different (*p* < 0.01).

| Compound | TS0-WR1 | TS50-WR1 | TS100-WR1 | TS0-WR2 | TS50-WR2 | TS100-WR2 |
|---|---|---|---|---|---|---|
| Sucrose (mg/100 g FW) | 3320 ± 380 b | 3422 ± 273 b | 4056 ± 321 a | 3421 ± 285 b | 3330 ± 301 b | 3730 ± 322 ab |
| Glucose (mg/100 g FW) | 4440 ± 420 ab | 4234 ± 413 b | 5256 ± 498 a | 4542 ± 376 ab | 4302 ± 412 b | 4545 ± 477 ab |
| Fructose (mg/100 g FW) | 4510 ± 314 a | 4443 ± 372 a | 5178 ± 503 a | 4476 ± 287 a | 4409 ± 442 a | 4430 ± 341 a |
| TSS (°Brix) | 12.6 ± 0.6 bc | 12.4 ± 0.5 c | 16.0 ± 0.7 a | 12.7 ± 0.5 bc | 14.7 ± 0.4 b | 13.7 ± 0.6 b |
| Ascorbic acid (mg/100 g FW) | 15 ± 1.0 c | 16 ± 0.9 bc | 30 ± 1.2 a | 16 ± 1.0 bc | 17 ± 0.9 b | 15 ± 1.0 c |
| Malic acid (mg/100 g FW) | 320 ± 30 b | 250 ± 25 c | 490 ± 43 a | 250 ± 32 c | 270 ± 28 bc | 230 ± 27 c |
| Citric acid (mg/100 g FW) | 1070 ± 90 c | 1070 ± 93 c | 2300 ± 98 a | 1140 ± 80 bc | 1210 ± 85 b | 1000 ± 87 c |

The observed overall mean total polyphenol content in the fruits was 650.72 mg GAE/100 g FW, with the highest value observed in fruits obtained from the TS50 GM (686.7 mg GAE/100 g FW) and WR1 irrigation regime (676.9 mg GAE/100 g FW). No statistical differences were observed among the GMs, WRs or their interaction. The highest DPPH activity (0.67% of inhibition) was observed in the fruits of the plants grown in TS100 and from the WR2 irrigation regime (0.63% of inhibition); however, no statistical variations were found among substrates, water regimes and their interactions. No correlation was found between the two nutraceutical variables (Figure 1).

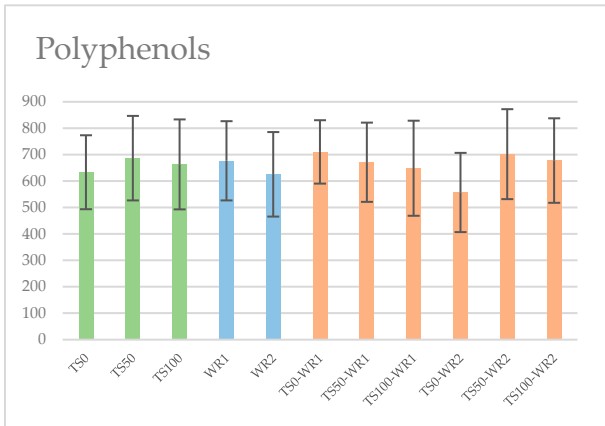 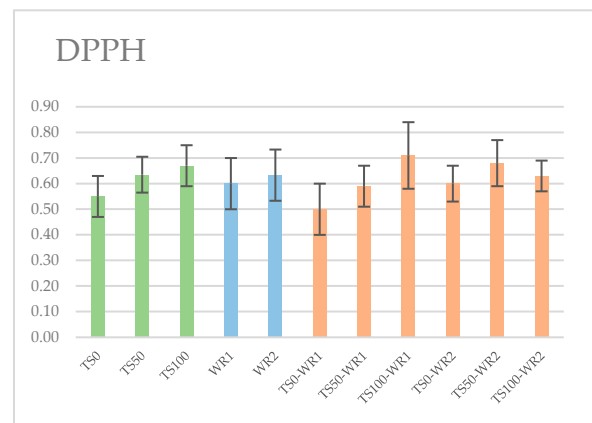

**Figure 1.** Total polyphenol content ((**left**); mg GAE equivalent/100 g FW) and DPPH radical-scavenging activity ((**right**); % of inhibition) observed in the fruits of the wild strawberry (*F. vesca*) plants grown in different growing media and under different irrigation regime.

### 3.3. Elemental and Organic Contaminant Analyses

Among the 21 different organic contaminants analyzed, none showed a concentration higher than the detection threshold of the applied methodology. Only the total heavy hydrocarbons (C10–C40) were revealed in quantities above the detection limits but at concentrations very close to the threshold (22 mg/kg FW) in fruits obtained from the plants grown in TS50-WR1 and TS50-WR2 (25.4 and 25.2 mg/kg FW, respectively) and in TS100-WR1 (25.3 mg/kg FW). The concentrations of Cd, Cr, Ni, Pb and Zn were found to be below the detection threshold (9.9 mg/kg FW) in all fruit samples, while P and Cu were present only in the fruits from the TS100-WR1 plants at concentrations of 19.0 mg/100 g FW and 0.18 mg/kg FW, respectively.

Table 4 shows the concentrations of the mineral elements and the statistical significance of the main factors and their interactions for calcium (overall average content 34.8 ± 3 mg 100 g FW), potassium (207 ± 2 mg/100 g FW), magnesium (17.6 ± 5 mg/100 g FW), sodium (3.5 ± 1 mg/100 g FW), iron (2.32 ± 0.4 mg/kg FW) and manganese (4.48 ± 0.6 mg/kg FW). In particular, considerable variation was observed among the growing media as well as the water regimes and their interactions. The TS0 (control) fruits revealed reasonably high concentrations of microelements such as Fe, Na, and Mn in addition to K. Otherwise, an increase in Ca was detected in TS50 (37.3 mg/100 g FW), showing a statistical value comparable to control's value. Regarding water regimes, the WR2 plants displayed a marked surge of K, Mg and Fe, while Ca and Mn were observed to be higher in the WR1 plants. The main factors and their interactions did not exert any effect on the fruit content of nitrogen.

### 3.4. Sensory Evaluation and Acceptance

The results from the PanelCheck sensory analysis are reported in Figure 2, which shows spider plots demonstrating the average values from the 10 panelists' scores, and biplots based on the principal component analysis are separately presented for each sensorial set. Regarding visual traits, the spider plot reveals high scores of fruit roundness and color uniformity in the TS100 fruit samples (for both WRs) when compared with the fruits obtained from the other GMs (Figure 2A, left). Moreover, the PCA biplot clearly shows a considerable separation of the fruit samples in the four quadrants on the basis of the different growing media (Figure 2A, right). No differences were detected between the two water regimes.

**Table 4.** Mineral element concentrations in the fruits of wild strawberry (*F. vesca* cv "Regina delle Valli") cultivated on three growing media with increasing amounts of treated port sediment (TS0, TS50 and TS100) under two irrigation regimes (WR1 and WR2). Mean separation within columns by Duncan's multiple range comparison test. Means followed by different letters are significantly different. ns = not significant; * significant at $p < 0.05$; ** significant at $p < 0.01$.

| Factor | Calcium (mg/100 g FW) | Potassium (mg/100 g FW) | Magnesium (mg/100 g FW) | Iron (mg/kg FW) | Sodium (mg/100 g FW) | Manganese (mg/kg FW) | Total N (% DW) |
|---|---|---|---|---|---|---|---|
| **GM** | | | | | | | |
| TS0 | 36.2 a | 2140 a | 17.1 a | 2.71 a | 3.9 a | 7.64 a | 2.20 ns |
| TS50 | 37.3 a | 2005 c | 17.0 a | 2.16 b | 3.6 b | 2.88 b | 2.10 ns |
| TS100 | 30.1 b | 2075 b | 17.5 a | 2.06 b | 2.9 c | 2.93 b | 2.45 ns |
| **WR** | | | | | | | |
| WR1 | 36.3 a | 2053 b | 17.0 b | 1.93 b | 3.8 a | 4.93 a | 2.18 ns |
| WR2 | 32.4 b | 2093 a | 18.3 a | 2.70 a | 3.7 a | 4.03 b | 2.32 ns |
| **GM × WR** | | | | | | | |
| TS0-WR1 | 38.4 a | 2190 a | 17.6 a | 1.96 c | 3.8 a | 8.86 a | 2.15 ns |
| TS0-WR2 | 33.2 c | 2090 b | 17.6 a | 3.46 a | 3.9 a | 6.42 b | 2.25 ns |
| TS50-WR1 | 38.2 a | 1870 d | 16.7 b | 1.96 c | 3.2 b | 2.79 d | 2.16 ns |
| TS50-WR2 | 35.4 b | 2140 b | 17.7 a | 2.36 b | 3.9 a | 2.98 c | 2.05 ns |
| TS100-WR1 | 32.4 c | 2100 b | 17.6 a | 1.86 c | 2.2 c | 3.15 c | 2.24 ns |
| TS100-WR2 | 29.3 d | 2050 c | 17.6 a | 2.26 b | 3.4 b | 2.71 d | 2.65 ns |
| **Significance** | | | | | | | |
| GM | ** | ** | * | ** | ** | ** | ns |
| WR | ** | ** | ** | ** | ** | ** | ns |
| GM × WR | ** | ** | ** | ** | ** | ** | ns |

The sensory evaluation resulted in relevant differences for the smell, taste and aftertaste parameters, which were attributed to the effect of the growing medium, while the irrigation regimes did not significantly affect the results (Figure 2B–D). The most relevant differences among the studied samples were related to some olfactory attributes, such as intensity and fruity smells, which were especially high in the TS50 samples (Figure 2B, left), as well as the graininess and sweetness taste attributes during chewing, which were substantially lower in the TS0 samples (Figure 2C, left). The profiles of the six samples were quite different for all the aftertaste attributes (Figure 2D, left), and the TS0 fruits in particular showed the lowest levels of fruity, floral, and vinous flavors in comparison with the other media (i.e., TS50 and TS100). All the PCA biplots obtained for the different attribute sets displayed a similar distribution pattern of samples, with a clear separation in relation to the three different growing media (Figure 2B–D, right). The PCA biplot obtained for all the sensorial attributes taken into account confirmed that the irrigation regime did not exert a relevant effect on the samples with respect to their sensorial attributes (Figure 3).

*3.5. Multi-Trait Approach*

In this case, a PCA was performed on the mean and standardized values of all the observed leaf pigment and fruit quality parameters. Firstly, a correlation analysis was performed to decrease the number of variables, which were removed according to a Pearson coefficient correlation greater than 0.95. In particular, 23 parameters were removed for having a high correlation with the selected traits (Figure S1). In more detail, floral and undergrowth smells, graininess and viscosity in chewing and floral and vinous flavors all demonstrated a significant relationship with overall satisfaction (r > 0.95). Other groups of leaf and fruit variables with r > 0.95 were carotenoids with sodium concentration; fructose, malic and ascorbic acids with citric acid; fruit roundness, vinous smell and fruity flavor with sweetness; fruit integrity and intensity of flavor with color uniformity; intensity of

spicy and off-flavor smells as well as spicy flavors and off-flavor taste with a fruity smell; sourness with color intensity; and herbaceous smell and bitterness with herbaceous flavor.

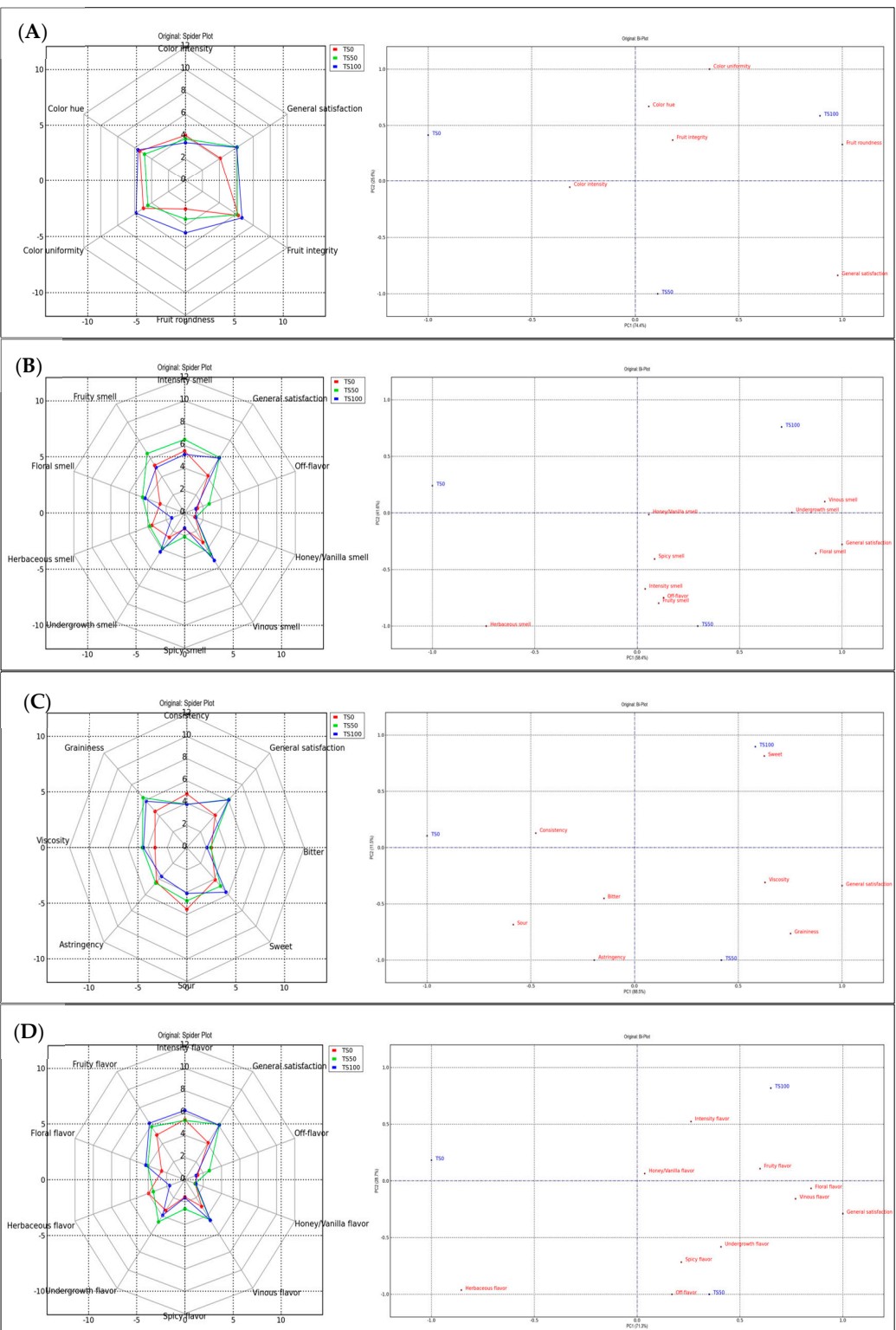

**Figure 2.** Sensory evaluation of wild strawberry (*Fragaria vesca* cv. "Regina delle Valli") fruits from plants cultivated in growing media differently enriched with remediated port sediments. (**A**) Spider plot and biplot of visual attributes on the entire fruit. (**B**) Spider plot and biplot of olfactory attributes of the entire fruit. (**C**) Spider plot and biplot of taste attributes during chewing. (**D**) Spider plot and biplot of aftertaste taste attributes.

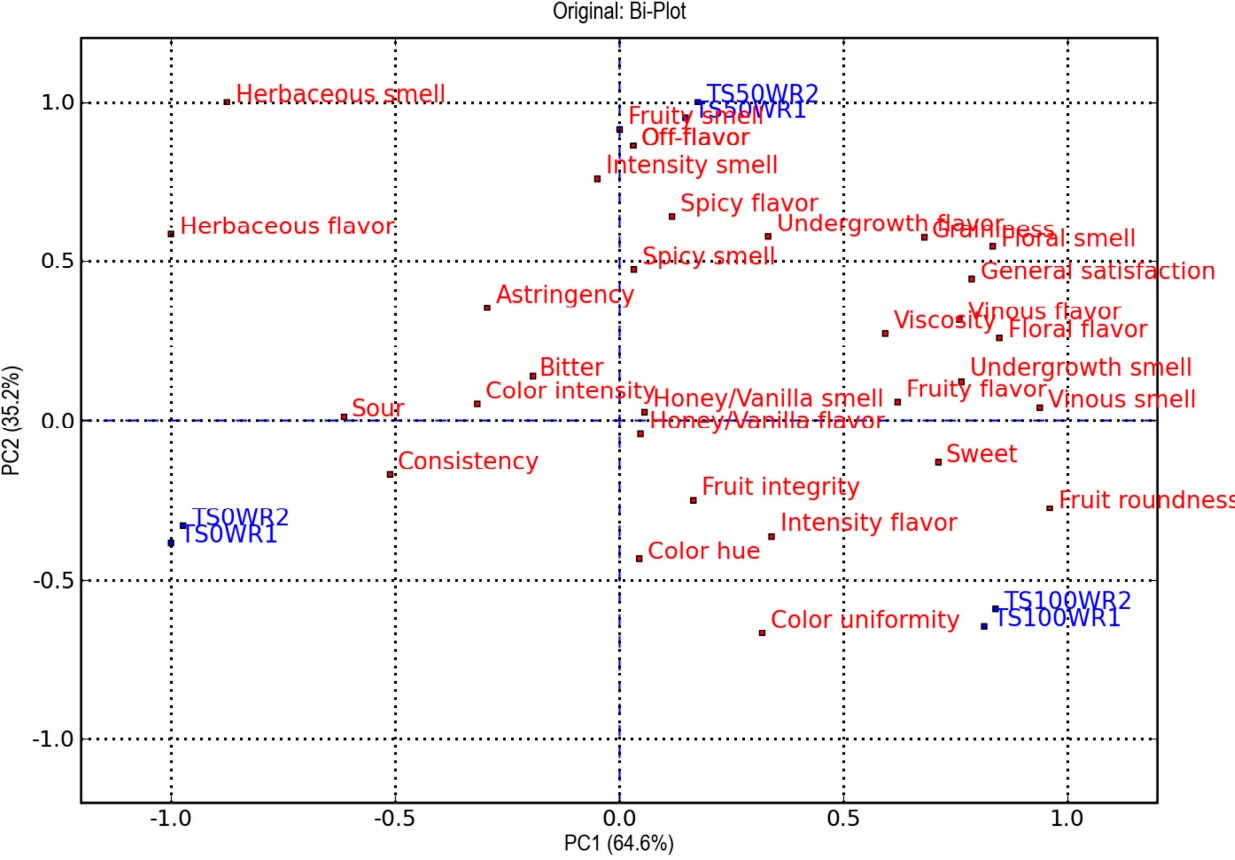

**Figure 3.** Sensory evaluation of wild strawberry (*Fragaria vesca* cv. "Regina delle Valli") fruits from plants cultivated in growing media differently enriched with remediated port sediments. Biplot of all sensorial attributes.

From these selections, a total of 29 response variables were used to obtain a PCA biplot (Figure 4). The three most relevant components (i.e., PC1, PC2 and PC3) described 83.62% of the total variation, divided into 44.17%, 22.38% and 17.07% respectively. PC1 was positively associated with fruit traits related to sweetness (i.e., taste sweetness, sucrose, glucose and TSS) and color uniformity; furthermore, it was negatively correlated with sensorial parameters (i.e., color intensity, chewing consistency, taste astringency and herbaceous flavor) and the chlorophyll a/b ratio as well as nutritional elements such as calcium and sodium (Table 5). Conversely, the second PC showed a reasonable relationship with sensorial descriptors referring to olfactory and aftertaste attributes (i.e., fruity and vanilla smells and undergrowth flavor) in addition to the overall satisfaction of the 10 assessors, while a negative correlation was revealed with fruit visual attributes (i.e., color hue and uniformity) as well as chewing consistency and manganese concentration. The resulting biplot displayed a visible separation of the samples according to the different sediment concentrations applied, while the water regimes did not affect differentiation inside the same GM. The two controls (TS0) were placed in the bottom-left part of the plot, resulting in a positive association with fruit consistency and manganese concentration. The TS50s samples were placed in the middle-top, evidencing a strong positive correlation with the PC2, in particular the fruity smell and undergrowth flavor factors. Finally, the TS100 samples were displayed in the middle right of the figure, showing a marked influence on PC1, specifically the sweetness traits.

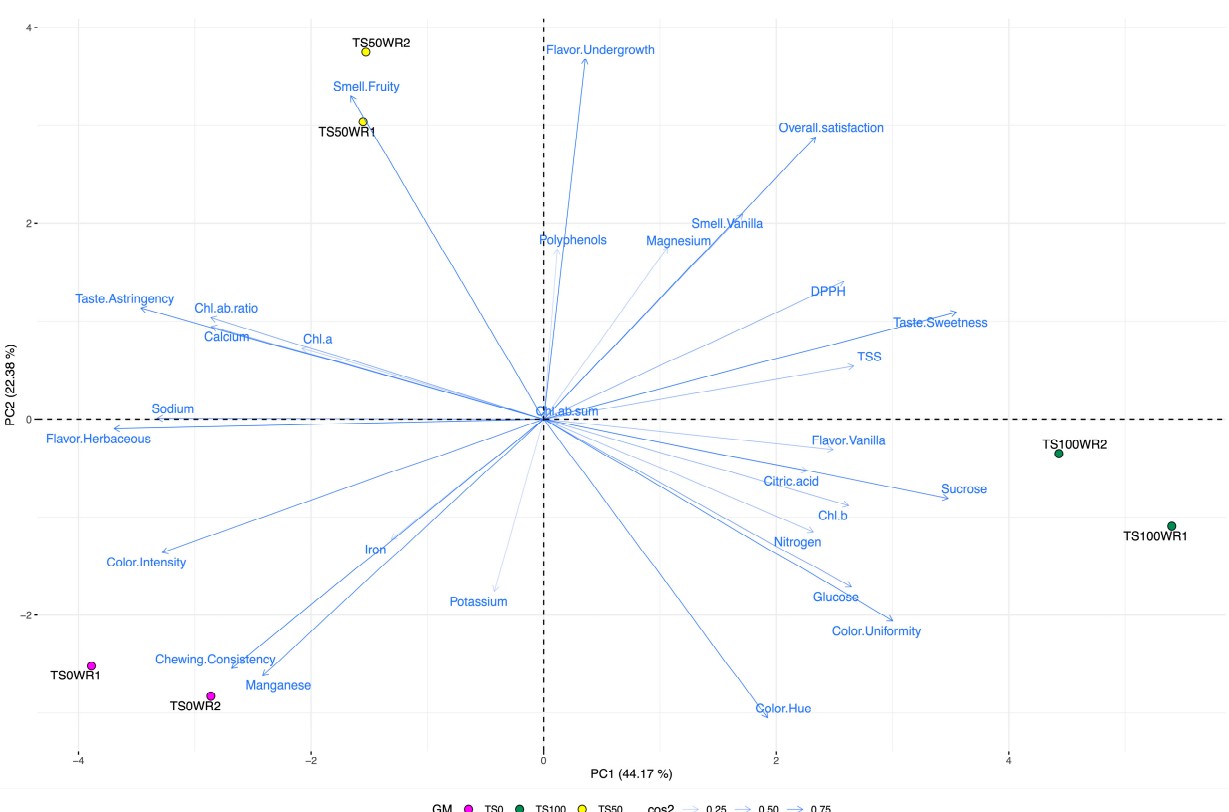

**Figure 4.** PCA biplot of 29 plant parameters selected from 52 original variables on the basis of Pearson correlation coefficient (r > 0.95) and showing the relationships among the six different treatments (GM×WR). The treatments are colored according to the three different GMs.

### 3.6. Correlation among Leaf Pigment, Fruit Quality and Substrate Parameters

Figure 5 shows a heatmap that was created by correlating all plant parameters with the entire growing media properties after plant cultivation (Table S1). This heatmap confirmed the findings of the PCA biplot based solely on plant parameters (Figure 4), grouping the six treatments (GM×WR) into three main clusters for GM, each one of which is subclustered by the water regime (WR). Additionally, the heatmap revealed intriguing relationships between plant/GM associations, dividing the studied parameters into four main clusters. The first group (A) was composed of different plant variables such as organic acids (i.e., malic, ascorbic and citric acids), sugars (i.e., sucrose, fructose, glucose and TSS), DPPH and substrate pH. The second cluster (B) was the largest (i.e., 25 variables), comprising different sensorial fruit traits (e.g., undergrowth, floral, vanilla and vinous smells; fruity, vanilla, floral and vinous flavors; overall satisfaction, graininess and viscosity chewing) and fruit elements (i.e., N and Mg), in addition to many GM macro- and microelements (i.e., ZN, Pb, Fe, Mg, Mn, Ni, $NO_3^-$ and P). The third cluster (C) was subdivided into several sub-clusters that mostly represented associations between some GM chemical properties (i.e., K, Na, TOC and TN) and fruit element contents (Fe, Ca, Na, K and Mn), in addition to some fruit-eating (i.e., astringency, bitterness, sourness and consistency) and visual attributes (i.e., color intensity) and herbaceous flavor and smell. Finally, the fourth group, which was the smallest (i.e., 13 variables), was composed of fruit olfactory and tasting parameters (i.e., spicy, fruity and off-flavor smells and smell intensity; undergrowth and spicy flavors; off-flavor taste), leaf pigment traits (i.e., Chl a and Chl a+b), total polyphenol content and growing media EC, and Cu and $NH_4^+$ concentrations.

**Table 5.** Component loadings of the selected plant variables according to the first two PCs. * Variables used to explain the PCs: considered threshold of 0.199. The percentage of variance per each PC and the cumulative percentage of variance are reported.

| Plant Variables | PC1 | PC2 |
|---|---|---|
| Color.Intensity | −0.246 | −0.143 |
| Color.Hue | 0.145 | −0.322 * |
| Color.Uniformity | 0.225 * | −0.217 * |
| Smell.Fruity | −0.125 | 0.348 * |
| Smell.Vanilla | 0.129 | 0.221 * |
| Chewing.Consistency | −0.202 * | −0.268 * |
| Taste.Astringency | −0.260 | 0.120 |
| Flavor.Herbaceous | −0.277 * | −0.010 |
| Flavor.Undergrowth | 0.027 | 0.388 * |
| Flavor.Vanilla | 0.187 | −0.032 |
| Taste.Sweetness | 0.267 * | 0.115 |
| Overall.satisfaction | 0.176 | 0.304 * |
| Chl.a | −0.156 | 0.076 |
| Chl.b | 0.197 | −0.093 |
| Chl.ab.sum | 0.005 | −0.002 |
| Chl.ab.ratio | −0.215 | 0.109 |
| Polyphenols | 0.009 | 0.183 |
| DPPH | 0.194 | 0.148 |
| Calcium | −0.215 * | 0.100 |
| Iron | −0.098 | −0.129 |
| Magnesium | 0.080 | 0.185 |
| Manganese | −0.181 | −0.276 * |
| Potassium | −0.032 | −0.186 |
| Sodium | −0.250 | 0.001 |
| Nitrogen | 0.174 | −0.121 |
| Sucrose | 0.261 * | −0.086 |
| Glucose | 0.199* | −0.181 |
| TSS | 0.200 | 0.057 |
| Citric.acid | 0.170 | −0.056 |
| Perc. of variance | 44.17% | 22.38% |
| Cum. Perc. of variance | 44.17% | 66.55% |

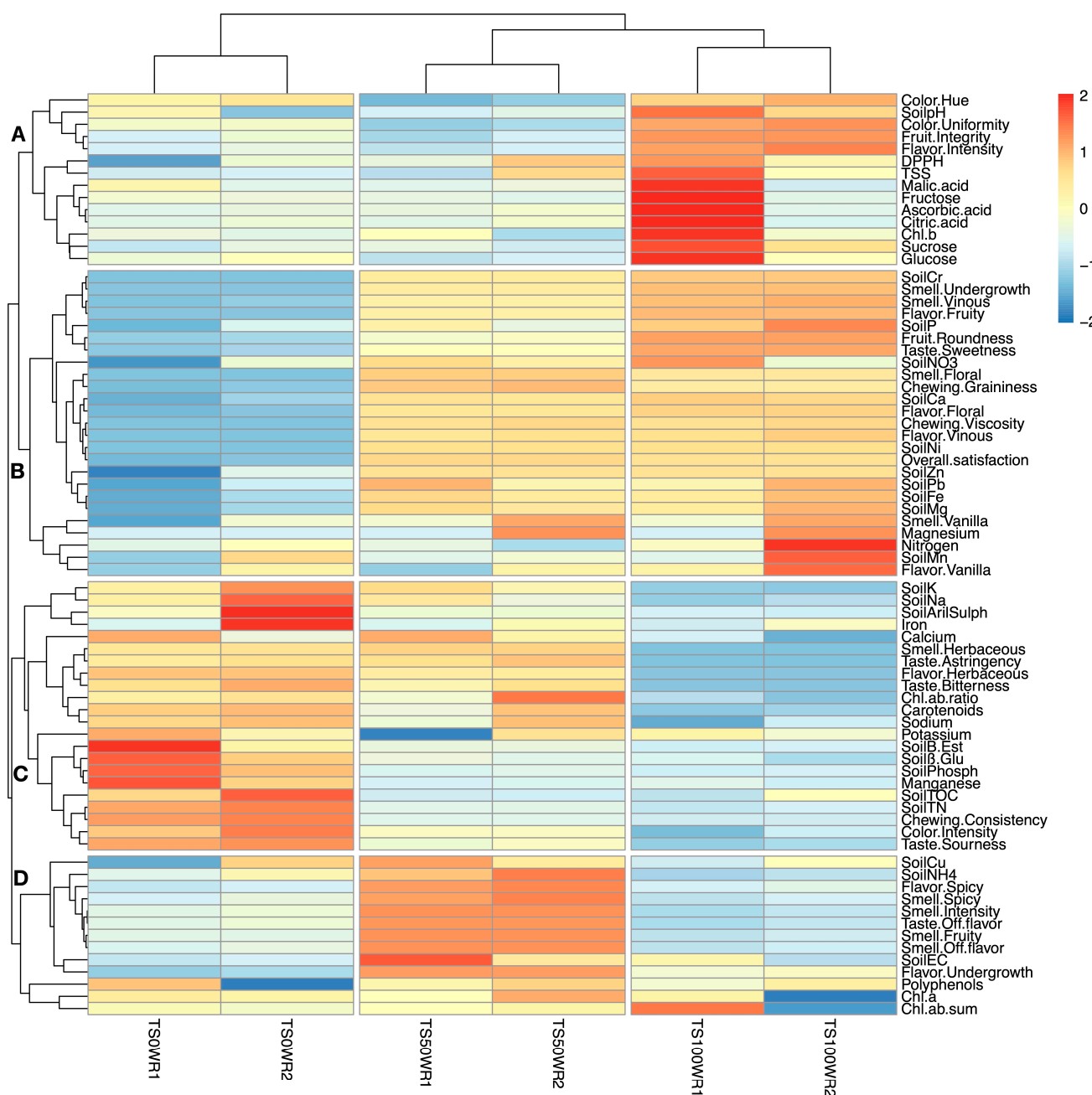

**Figure 5.** HeatMap based on the average values of the entire set of plant parameters of *Fragaria vesca* cv "Regina delle Valli" fruits in addition to final GM chemico-physical properties of the six GM×WR applied treatments. The four main clusters obtained among variables are indicated with capital letters (i.e., A, B, C, and D).

## 4. Discussion

Experimental studies dealing with wild strawberry (*F. vesca* L.) cultivation in different growing media are very limited; similarly, the literature on experimental results related to the quality of wild strawberry fruit is poor. For the sake of clarity, the observed results will not be compared to those observed for strawberry (*Fragaria × ananassa*) unless particularly specified.

### 4.1. Leaf Pigment Analysis

Our results clearly show that the leaf carotenoid content, in contrast with the chlorophyll concentration, was significantly affected by the growing media (GM) composition

(Table 2), with values almost twofold in TS0 and TS50 when compared to TS100. Our overall average content (21.66 μg/mg FW) was about 20-fold higher than those observed by Lema-Rumińska et al. [15] in Poland; this discrepancy may be attributed to the different cultivars and, to a major extent, to the different growing temperatures (a maximum temperature of about 30 °C in our trial against 15 °C in the Polish experiment), and to the physico-chemical properties of the substrate.

### 4.2. Nutritional and Nutraceutical Analysis

The observed values for sugar content (Table 3) are in agreement with those observed by Caruso et al. [32] and Doumett et al. [13] in "Regina delle Valli" fruits obtained via the Nutrient Film Technique (NFT) cultivation system and field cultivation in different Italian locations, respectively. Accordingly, the ratios between the monosaccharides of the different treatment results are comparable to those reported by the literature. Similarly, comparable values of soluble solids in the juice from the berries were found in our study, ranging from 12.4 to 16.0 °Brix vs. the 9.5 to 15.9 °Brix observed in other studies, with the latter data related to experiments conducted in different growth environments. In addition to sugars, organic acids are a relevant component of fruit quality in terms of both antioxidant activity and sensorial quality. The values observed in our study are fully comparable with those observed by Doumett et al. [13] and Del Bubba et al. [11] on fruits grown in the open air and under tunnels in Italy. The organic acid contents were comparable to those observed by Caruso et al. [33] in the hydroponic cultivation of the "Regina delle Valli" cultivar, with a higher concentration of citric acid than malic acid. In our study, we found remarkable levels of ascorbic, citric and malic acid in the TS100-WR1 harvested fruits. These values were twice as high as the average values observed in the other treatments and reported by the previously cited authors. Moreover, we found a total polyphenol content of about 650 mg GAE equivalent/100 g FW, with relatively small variations between the treatments. Such a value is comparable to those observed in *Fragaria × ananassa* fruits by Doumett et al. [13] and Zahid et al. [34] and by Del Bubba et al. [11] in cultivated *F. vesca* fruits.

### 4.3. Elemental and Organic Contaminant Analyses

The presence of organic contaminants in the fruits obtained from the plants grown in all tested growing media was negligible, revealing concentration values lower than the thresholds indicated by European food safety regulations. Similar results were obtained by Tozzi et al. [8,9] in strawberries of *Fragaria × ananassa* and in lettuce in the same substrates, TS50 and TS100, confirming that the maritime-treated sediments used as media in this study are suitable for agricultural production. To the best of our knowledge, the literature on the mineral content of cultivated wild strawberry fruits is very limited. Taking into account that the macro- and micronutrient contents showed significant differences due to the treatments in our study, the concentrations of calcium, potassium and magnesium are comparable to the values reported by Jurgiel-Małecka et al. [35] and Caruso et al. [33] in fruits from the "Rugia" and "Yellow Wonder" wild strawberry cvs. cultivated in Poland and in the fruits of "Regina delle Valli" obtained via hydroponics, respectively. Similarly, micronutrient concentrations (i.e., iron, manganese and copper) were found to be comparable to the amounts reported in the two cited papers. Overall, the mineral element concentrations were comparable to those reported by Aslantas et al. [36] in fruits of *Fragaria vesca* plants grown in the wild in the Northeast Anatolia Region of Turkey. In our study, the fruit sodium content was significantly affected by the growing media and water regime and their interaction. Indeed, the highest Na concentration (3.9 mg/100 g FW) was observed in the TS0-WR2 and TS50-WR2 fruits, demonstrating that TS100 had no direct effect on the sodium content.

### 4.4. Sensory Evaluation and Acceptance

Sensory evaluation had been previously applied to "Alpine" and "Regina delle Valli" wild strawberry fruits obtained via different cultivation systems [11]. Our research method-

ology enabled us to differentiate the fruits based on the applied treatments using various sensory parameters. Interestingly, we found that the growing media had a more significant impact on fruit quality than the irrigation regime. Fruits from the TS100 treatment consistently received the highest scores for flavor regardless of the irrigation regime.

### 4.5. Correlation among Leaf Pigment, Fruit Quality and Substrate Parameters

The relationships between the plant parameters and the soil chemico-physical properties are complex and not easy to interpret. Multivariate approaches can be useful for associating and developing some considerations that may, even if partially, explain the observed results. As shown in Figure 5, notwithstanding a complex pattern of clustering, four main groups are observable among the plant and GM parameters, each one including both types of characteristics and hence showing a quite significant number of high correlation values, which are reported in Table S2 for all the pairwise variables.

The smallest cluster (Cluster D; Figure 5) holds a number of fruit olfactory and tasting parameters linked to the soil EC, and Cu and $NH_4^+$ contents, while the contents of chlorophyll a and a + b (particularly high in leaves of plants grown on TS100) are inversely correlated to the total polyphenol content. This association was also observed by Lin et al. [37] in a study on *Camellia sinensis* cultivated on different substrates and by Sarker and Oba [38] in selected cultivars of *Amaranthus tricolor* grown in pots with different irrigation regimes. A similar performance was noted by Khosropour et al. [39] on a *Berberis integerrima* cultivation using vermicompost and biochar to alleviate cadmium stress. A strong correlation ($r > 0.89$) was observed between the $NH_4^+$ content in the soil (which was found to be particularly high in TS50 GMs) and a spicy flavor, spicy and fruity smells, and the intensity of smell and off-flavor. Such an unexpected association can be broadly related to the modifications exerted by different types of nitrogen sources and metabolisms on gene expression, as recently observed by Wang et al. [40] in tea plants. Nevertheless, further studies are necessary to corroborate this hypothesis.

Regarding cluster C (Figure 5), the high values of TN, TOC, K and Na concentrations and the soil enzymatic activity observed in the TS0 media convey an increase in fruit color intensity, consistency, sourness, bitterness and astringency and the concentrations of mineral elements such as Mn, K, Na and Ca. Aside from the contribution of each soil parameter, it is well-known that the chemical composition of the soil, substrates and fertilization exert a relevant effect on fruit quality, as demonstrated, for example, on wild strawberries [11] and *Fragaria × ananassa* cultivars [41]. The correlation between soil enzymatic activity and the quoted fruit parameters ($0.68 < r < 0.96$) is not easily explainable in terms of a cause/effect mechanism; nevertheless, the enzymatic activity observed in the peat growing media with the richest peat concentration was found to be higher than in the TS50 and TS100 substrates, confirming a generally different balance of elements and chemical conditions which influence microorganisms and plants [42]. Moreover, the heatmap proved the results regarding the fruit mineral element contents, confirming the absence of a relationship between the TS100 media and the amount of Na in wild strawberry fruits. Table S2 shows a noteworthy and very close correlation (with $r > 0.95$) between the concentration of Mn in fruits and GM enzymatic activity (specifically, ß-glucosidase, butyrate esterase and phosphatase activity). This association can be attributed, at least in part, to the impact of microorganism activity in the soil, which can alter the availability and absorption of nutrients in plants, as discussed by Mohamed et al. [43]. An interesting correlation was identified regarding the fruits' Na concentration and the carotenoid ($r = 0.97$) and chlorophyll a/b ratios ($r = 0.87$) in the leaves. Carotenoids fulfil various vital functions in plants, and they are an important component required for light harvesting and photoprotection. Carotenoids act also as oxidative stress "sensor" and "signal" upon oxidation, which is an integral component of several stress conditions in plants related to biotic and abiotic environmental challenges [44]. It is possible to speculate that the chemico-physical characteristics of the TS0 medium may trigger a higher concentration of carotenoids in the leaves of plants, which may also lead to an accumulation of Na in

the fruits. This phenomenon could be an indication of a stress factor that alters the ratio between chlorophyll a and b in plants. Ullah et al. [45] found a positive correlation between Na content in mungbean (*Vigna radiata* L.) plants and carotenoids when artificially increasing the growing substrate salinity.

Cluster B includes 25 parameters (namely, 10 related to the growing media and 15 related to plant characteristics). In addition to a strong association (r~0.95) of clustered soil element contents (i.e., Zn, Pb, Mg and Fe), a sub-group showed a close positive correlation between the Mn concentration in the GMs and the total nitrogen and magnesium contents and the vanilla smell and taste of the fruit (r~0.82), with the highest values observed for the TS100-WR2 treatment, thus confirming the suitability of the remediated marine sediment for wild strawberry production. The overall sensorial fruit satisfaction results were closely and positively correlated (r > 0.90) with other fruit parameters (floral and vinous smell, floral taste, chewing graininess and viscosity), with nickel and calcium soil contents, and to $NO_3^-$ (r = 0.76), with the highest values found for the TS100-WR1- and TS100-WR2-treated plants. A balanced availability of nickel, calcium and $NO_3^-$ in the substrate has been reported to exert a positive effect on the sensory qualities of strawberry (*Fragaria* × *ananassa*) fruits [46–50]. Another interesting sub-cluster holds soil concentrations of Cr and P with the sweetness taste and fruity and other smell traits (i.e., vinous and undergrowth), with r > 0.90. The taste sweetness results were also highly and positively correlated with the soil contents of Ca, Mg, Fe, Ni and Zn (Table S2), thus demonstrating the relevance of mineral element concentrations and their ratios on an important fruit quality trait: the perception of sweetness [51].

In cluster A, organic acids, sugars, TSS and DPPH were grouped together, revealing the highest values in fruits obtained from plants grown in TS100 under WR1. The findings of our study align with the PCA biplot analysis for all fruit quality parameters, indicating that the TS100s were strongly impacted by sugar content and citric acid (as shown in Figure 4). Additionally, previous research has shown that organic acids play a significant role in determining the sensory characteristics of fruit, particularly with respect to strawberry flavor [52,53]. Our study corroborates these findings, as the intensity of flavor was grouped with ascorbic, malic and citric acids in cluster D. No linear association was noticed between organic acids and DPPH versus the polyphenol fruit content, while the total soluble solids (TSSs) of the fruits (overall mean 13.7 °Brix) were positively correlated with sucrose, glucose and fructose (r > 0.72).

## 5. Conclusions

Technological innovation in agriculture, an improvement in farmers' skills and climate change are leading to relevant transformations of cropping systems by pushing toward a transition from open-field cultivation to soilless greenhouse cultivation. The wild strawberry is a paradigmatic example of this, as soilless cultivation allows us to improve, among others, the efficiency of fruit harvesting, sorting and packaging, thus augmenting the fruits' commercial attractiveness. Cultivation out of soil and in tunnels also makes it possible to increase the number of production cycles. However, soilless cultivation requires the use of substrates based on peat or coconut fibers, which have a high environmental impact. The use of a medium obtained from the remediation of marine port sediments can therefore represent an interesting solution that responds to the needs of food safety and the commercial offer of fruits of a high sensorial and nutraceutical quality, such as those obtained in the present study. This study demonstrates that remediated marine port sediments can be used in the production of safe *Fragaria vesca* fruits which show high-quality parameters in terms of their sensorial attributes and nutraceutical value. Furthermore, our results serve as inspiration for further research into the impact of soil chemical and physical properties on fruit quality. This area of study is highly complex and remains inconsistently explored.

**Supplementary Materials:** The following supporting information can be downloaded at: https://www.mdpi.com/article/10.3390/su15097338/s1, Figure S1: Correlation plot on 52 plant variables according to Pearson correlation coefficient; Table S1: Chemico-physical properties of growing substrates analyzed at the final stage of plant cultivation in relation to water regime; Table S2: Pearson correlation coefficient (r) between the studied variables of plants and chemico-physical properties of substrates at the end of plant cultivation.

**Author Contributions:** Conceptualization, E.G., G.M. and S.N.; methodology, S.N.; software, E.G., L.B., D.B. and S.N.; validation, E.G., G.M. and S.N.; formal analysis, E.G., L.B., D.B. and S.N.; investigation, D.B., W.A.P. and G.C.; resources, S.N.; data curation E.G., L.B., D.B., W.A.P., G.M., G.C. and S.N.; writing—original draft preparation, E.G., L.B., D.B. and S.N.; writing—review and editing, E.G., L.B., D.B., W.A.P., G.M., G.C. and S.N.; visualization, E.G., L.B., D.B.; funding acquisition, S.N. All authors have read and agreed to the published version of the manuscript.

**Funding:** This research was supported by European Life Project SUBSED "Sustainable substrates for agriculture from dredged remediated marine sediments: from ports to pots" (LIFE 17 ENV/IT/000347).

**Institutional Review Board Statement:** Not applicable.

**Informed Consent Statement:** Not applicable.

**Data Availability Statement:** The data are contained within the article or Supplementary Materials.

**Conflicts of Interest:** The authors declare no conflict of interest.

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
