# Peer review of "Effect of Innovative Sediment-Based Growing Media on Fruit Quality of Wild Strawberry (Fragaria vesca L.)"

_sustainability, doi:10.3390/su15097338_

Round 1

Reviewer 1 Report

The study is very well planned and is a valuable study that reveals the importance of waste in the evaluation of soilless agricultural environment.  However, adding important quality parameters such as yield and fruit size and weight to the study will make the study more valuable and will be beneficial for both researchers and producers. In addition to these, the study is eligible for publication if the following corrections are made.

On line 21, the details of the water regime should be written in parentheses.

Line 21 main factor should be specified

In the summary section, the results are given very superficially and should be summarized in more detail.

In the material and method part, parameters such as yield, fruit weight, and length should not be added.

Line 212-214 Although it is said that it is not statistically significant, there are significant ones in the table. paragraph should be arranged according to the table.

Line 225. Title 3.2 

There seem to be differences in sugar content between treatments, but statistical analysis has not been done. Why ? You already have duplicate data. It would be more appropriate to update this section by performing statistical analysis.

Line 340-363 and Line 396-427 Arrange the paragraph shifts to the left of the figures. 

Title Line 436 and Line 488 Discuss with more publications

The parts containing the findings in figure 5 and below should be included in the findings section. These issues were not mentioned in the findings section. Separate chapter 4.5 as findings and discussion and evaluate them separately.

I think that soilless agriculture studies are very important and valuable. I wish you success in your work.

Author Response

The study is very well planned and is a valuable study that reveals the importance of waste in the evaluation of soilless agricultural environment. However, adding important quality parameters such as yield and fruit size and weight to the study will make the study more valuable and will be beneficial for both researchers and producers. In addition to these, the study is eligible for publication if the following corrections are made.

We thank the reviewer for his valuable comments and suggestions.

As regards specifically the request of adding values concerning production parameters such as yield, fruit size and fruit weight, we would like to point out that these parameters have not been unambiguously included in this work.

In fact, the present experimentation on soilless wild strawberries cultivation on sediment-based substrates under different water regimes has produced a very large number of results. Therefore, it was decided to divide the results into two very separate and distinct papers also in terms of topics and interests for the readers: one essentially concerning yield and productivity; the other focused essentially on the quality of the production.

The first paper, presented at the IHC 2022 Conference in Angers, therefore reported production values such as crown diameter, shoot length, leaves per plant, Spad index, chroma index, leaf area, plant and total yield, fruit size, fruit fresh weight and biomass root and above ground parameters. For fairness and integrity, it was therefore decided not to use any of these parameters even in the multivariate analysis of the present paper submitted for Sustainability.

We had not included the reference of the above-mentioned paper that is being published in Acta Horticulturae within 60 days, although already accepted, since not yet published and printed. However, we have now realized that it is also possible to insert an article that has not yet been published and, therefore, if you deem it appropriate, we could add a reference to our previous paper on F. vesca productivity in the introduction section, as follows:

Prior studies have established that the composition of the growing media and irrigation practices can impact the yield, fruit quality, and various morphological, phenological, and organoleptic parameters of wild strawberries [9, 11]. In this context, a previous work [16], based on the same experiment reported below, was focused on the productivity of F. vesca ‘Regina delle Valli’ demonstrating that the growth and yield of plants cultivated on sediment-based growing media were comparable or superior to those grown on commercial peat. Specifically, in this research paper we investigate how remediated sediment-based growing substrates and various water regimes affect ‘Regina delle Valli’ fruit quality traits, such as nutraceutical properties and sensorial parameters.

On line 21, the details of the water regime should be written in parentheses.

The details of the water regime have been added.

Line 21 main factor should be specified

The main factors have been specified.

In the summary section, the results are given very superficially and should be summarized in more detail.

Specific details about the results have been added in the abstract.

In the material and method part, parameters such as yield, fruit weight, and length should not be added.

Parameters of plant yield, fruit size and weight have not been included in the materials and methods, since have been reported in a previous paper presented at the IHC 2022 Conference.

Line 212-214 Although it is said that it is not statistically significant, there are significant ones in the table. paragraph should be arranged according to the table.

The paragraph has been modified and arranged according to table results.

Line 225. Title 3.2

There seem to be differences in sugar content between treatments, but statistical analysis has not been done. Why ? You already have duplicate data. It would be more appropriate to update this section by performing statistical analysis.

The statistical analysis has been performed and letters indicating statistically significant variations have been added.

Line 340-363 and Line 396-427 Arrange the paragraph shifts to the left of the figures.

Sorry, we don't understand what is being asked.

Title Line 436 and Line 488 Discuss with more publications

In line 436 more publications have been inserted to improve the discussion. As regards the references required for line 488, we believe that the five cited references are appropriate

The parts containing the findings in figure 5 and below should be included in the findings section. These issues were not mentioned in the findings section. Separate chapter 4.5 as findings and discussion and evaluate them separately.

Findings reported in Figure 5 have been separately reported and discussed in chapters 3.6 (Results) and 4.5 (Discussion).

I think that soilless agriculture studies are very important and valuable. I wish you success in your work.

We thank the reviewer for appreciating our research.

Author Response

Dear editor

Thank you for inviting me to evaluate the manuscript entitled “Effect of innovative sediment-based growing media on fruit quality of wild strawberry (Fragaria vesca L.)”. In my opinion, some contents of this paper need to be improved.

1. Line 80-84: The physical and chemical properties of commercial peat-based substrate and the remediated sediment should be presented in this part.

The physico-chemical properties of the growing media before cultivation start have been reported (Table 1).

2. Line 82: Was volume ratio appropriate for this experiment here?

The percentages of sediment used were chosen on the basis of previous experiments carried out on strawberries and lettuce. Certainly, intermediate sediment percentages of 25% and 75% could also be chosen and included in the experimental design, but we didn't have the possibility to include so many different treatments. We preferred to use only the 50% sediment percentage and favor 2 different water supplies.

3. Table 1: The standard deviation should be added in this table.

Standard deviations have been added in the previous Table 1, now Table 2.

4. Table 2: Lowercase letters that representing the significance of the difference should be added to this table.

Lowercase letters have been added.

5. Figure 1: Lowercase letters that representing the significance of the difference and standard deviation should be added to this figure.

Figure 1 has been modified according to the reviewer’s suggestion.

6. Table 3: Many of the letters were mislabeled. Please check the table carefully.

Letters have been checked and properly modified.

7. “4. Discussion”: The authors should not classify the discussion according to different indicators, but should integrate all indicators for in-depth discussion.

The discussion was divided into opening short main topics for greater clarity and simplicity of reading, as is often proposed in various scientific articles. Then the last and very extensive section of the discussion was dedicated to an in-depth integrated analysis of all the indicators as a whole.

Round 2

Reviewer 1 Report

The authors responded appropriately to the referee's comments and cited their publications by specifying the parameters they could not add. The study is eligible for publication in this edited form. Good luck to the researchers.